# PCR-Based Microarray Enhances Diagnosis of Culture-Negative Biopsied Tissue in Patients with Invasive Mold Infections: Real-World Experience in a Tertiary Medical Center

**DOI:** 10.3390/jof10080530

**Published:** 2024-07-29

**Authors:** Hao-En Jan, Chin-Shiang Tsai, Cong-Tat Cia, Ching-Chi Lee, Ying-Wen Chen, Nan-Yao Lee, Chia-Wen Li, Ming-Chi Li, Ling-Shan Syue, Ching-Lung Lo, Tsung-Chain Chang, Chi-Jung Wu, Wen-Chien Ko, Po-Lin Chen

**Affiliations:** 1Department of Internal Medicine, National Cheng Kung University Hospital, College of Medicine, National Cheng Kung University, Tainan 701, Taiwan; david12030920@gmail.com (H.-E.J.); jasonmammal@gmail.com (C.-S.T.); ctcia@mail.hosp.ncku.edu.tw (C.-T.C.); nanyao@mail.ncku.edu.tw (N.-Y.L.); li.cw29@gmail.com (C.-W.L.); asurangle@gmail.com (M.-C.L.); lingshang03@gmail.com (L.-S.S.); raycigol@gmail.com (C.-L.L.); winston3415@gmail.com (W.-C.K.); 2Institute of Clinical Medicine, College of Medicine, National Cheng Kung University, Tainan 701, Taiwan; 3Department of Internal Medicine, National Cheng Kung University Hospital Douliu Branch, College of Medicine, National Cheng Kung University, Yunlin 640, Taiwan; didiowen@gmail.com; 4Center for Infection Control, National Cheng Kung University Hospital, Tainan 704, Taiwan; chichingbm85@yahoo.com.tw (C.-C.L.); wu.chijung@msa.hinet.net (C.-J.W.); 5Clinical Medical Research Center, National Cheng Kung University Hospital, Tainan 704, Taiwan; 6Department of Medical Laboratory Science and Biotechnology, College of Medicine, National Cheng Kung University, Tainan 701, Taiwan; tsungcha@mail.ncku.edu.tw; 7National Institute of Infectious Diseases and Vaccinology, National Health Research Institutes, Tainan 704, Taiwan; 8Diagnostic Microbiology and Antimicrobial Resistance Laboratory, National Cheng Kung University Hospital, Tainan 704, Taiwan; 9Department of Microbiology and Immunology, College of Medicine, National Cheng Kung University, Tainan 701, Taiwan

**Keywords:** polymerase chain reaction, microarray, mold infection, diagnostic performance

## Abstract

A fungal polymerase chain reaction (PCR) amplifies conserved genes across diverse species, combined with the subsequent hybridization of amplicons using a specific oligonucleotide microarray, allowing for the rapid detection of pathogens at the species level. However, the performance of microarrays in diagnosing invasive mold infections (IMI) from infected tissue samples is rarely reported. During the 4-year study period, all biopsied tissue samples from patients with a suspected IMI sent for microarray assays were analyzed. A partial segment of the internal transcribed spacer (ITS) region was amplified by nested PCR after DNA extraction. Amplicons were hybridized with specific probes for a variety of mold species using an in-house oligonucleotide microarray. A total of 80 clinical samples from 74 patients were tested. A diagnosis of an IMI was made in 10 patients (4 proven, 1 probable, 3 possible, 2 clinical suspicion). The PCR/microarray test was positive for three out of four proven IMIs, one probable IMI, and one out of three possible IMIs. Two patients with positive PCR/microarray findings were considered to have clinical suspicion of an IMI, and their responsible physicians initiated antifungal therapy despite the absence of supporting microbiological and histological evidence. Clinical diagnoses were categorized into non-IMI and IMI groups (including proven, probable, possible, and clinical suspicion). The sensitivity and specificity of the microarray in diagnosing the IMIs were 70% and 95.7%, respectively, while the sensitivity and specificity of the culture and histological findings were 10%/96.3% and 40.0%/100%, respectively. PCR-based methods provide supportive microbiological evidence when culture results are inconclusive. The combination of a microarray with fungal culture and histology promotes the precise diagnosis of IMIs in difficult-to-diagnose patients.

## 1. Introduction

Invasive mold infections (IMI) affect a variety of immunocompromised hosts and are associated with high crude mortality and economic burden [1]. These infections present a significant clinical challenge due to their diagnostic difficulties, particularly in cases where traditional culture methods yield negative results. Correct species identification, which guides adequate antifungal therapy, is a critical step in saving patients with IMIs. Despite advancements in antifungal therapies, timely and accurate diagnosis remains crucial for optimizing patient outcomes. Conventional diagnostic approaches for IMIs often rely on culture-based methods, which have limitations, including low sensitivity and delayed results. Additionally, the histopathological examination of tissue samples can provide valuable information although it may sometimes be difficult to assess. However, phenotypic identification of less frequently encountered opportunistic molds poses special problems because routine microbiological laboratories may lack the expertise, equipment, or technologies required [2].

Molecular techniques based on polymerase chain reaction (PCR) and sequencing technology allow laboratories to accurately identify pathogens at the species level based on the determination of sequences of the conserved genes for pathogens. The partial segment of the internal transcribed spacer (ITS) region, comprising partial 18S rRNA, ITS1, 5.8S rRNA, ITS2, and partial 28S rRNA, help differentiate fungal species [3,4]. In contrast to “untargeted” genome sequencing methods, the DNA microarray (or DNA chip) assay has proven to be a valuable tool for the rapid diagnosis of a variety of microorganisms at the same time [5,6,7,8,9,10]. An array of oligonucleotide probes that have been designed based on specific ITS regions can detect clinically important fungi [3,4,11,12,13,14,15].

In our previous studies, a wide spectrum of in-house oligonucleotide arrays had been validated to identify clinical isolates of bacteria, mycobacteria, molds, dermatophytes, and yeasts [3,4,7,9,10,11,12,13,16,17,18,19,20]. DNA microarray platforms in the diagnosis of fungal infections have been proven achievable [14,16,21,22]. In a study conducted by Kuo et al., the laboratory developed a dot hybridization assay that could diagnose *Aspergillus* and *Fusarium* keratitis rapidly and with excellent sensitivity and specificity [23]. Despite their potential benefits, the clinical utility of PCR-based microarrays in diagnosing IMIs from culture-negative biopsied tissue remains relatively underexplored and lacks real-world data validating the clinical performance. In the present study, we reviewed the PCR-based microarray assay and its performance in diagnosing IMIs at our institution.

## 2. Materials and Methods

### 2.1. Setting

The study was conducted at a 1300-bed tertiary medical center in Southern Taiwan. Between January 2017 and December 2020, microarray analysis was performed on biopsied tissues from patients with suspected IMIs at the hospital-affiliated diagnostic laboratory. Clinical features, including demographic factors, clinical diagnosis, microbiological cultures, and histological results, were recorded. The study was approved by the Institutional Review Board (IRB), National Cheng Kung University Hospital (NCKUH) (A-ER-110-039).

### 2.2. DNA Extraction and ITS Amplification for Mold

DNA was extracted from fresh biopsied tissues with the DNeasy^®^ Blood and Tissue kit (Qiagen, Hilden, Germany) according to the manufacturer’s instructions. The ITS genome of extracted DNA was amplified by nested PCR as previously described [16]. The first amplification was conducted using the universal primer pair, V9D (5′-TTAAGTCCCTGCCCTTTGTA-3′) and LS266 (5′-GCATTCCCAAACAACTCGACTC-3′).

PCR was performed with 2.5 μL of the template DNA diluted in a total reaction volume of 25 mL, consisting of 1× PCR buffer, 1.5 mM MgCl_2_, 1U of HotStarTaq DNA Polymerase (Qiagen, Hilden, Germany), 200 μM of each deoxynucleoside triphosphate (dATP, dGTP, dCTP, and dTTP), and 0.7 mM of each primer. The thermocycling conditions used for the first run of PCR were as follows: initial denaturation, 94 °C for 10 min; followed by 25 cycles of denaturation at 95 °C for 1 min, annealing at 60 °C for 1 min, and extension at 72 °C for 1 min. The final extension step was at 72 °C for 7 min.

A total of 1 μL of the product from the first reaction was then used as the DNA template for the second run of PCR using the universal primer pair, ITS1-digoxigenin (5′-digoxigenin-TCCGTAGGTGAACCTGCGG-3′) and ITS4-digoxigenin (5′-digoxigenin-TCCTCCGCTTATTGATATGC-3′) [24], which aimed to amplify a conserved portion of the 5.8S ribosomal DNA, the intervening ITS2, and a small part of the 28S ribosomal DNA. The reaction mixture, with a total volume of 50 μL, was utilized in the second run of PCR under the same conditions as those for the first run. A negative control was included in each test run by replacing the template DNA with sterile water in the PCR mixture. A positive control was obtained using DNA extracted from *Exophiala dermatitidis* ATCC 38714.

### 2.3. Fabrication of Membrane Assays and Hybridization Procedures

Clinically important molds detected by species-specific oligonucleotide probes were referred to and modified from our previous research and are listed in Appendix A [3,4,16,17]. The mold array included 78 dots for species identification, 1 positive control, 1 negative control, and 19 position markers. The arrangement of species-specific oligonucleotide probes for molds (Appendix A) and the processes of oligonucleotide hybridization in the DNA microarray are illustrated (Appendix A).

The oligonucleotide probes (10~20 μM) were drawn into wells of 96-well microtiter plates and spotted onto a positively charged nylon membrane measuring 1.1 × 0.8 cm (Roche, Mannheim, Germany) by an Ezspot arrayer (SR-A300; EZlife Technology, Taipei, Taiwan) using a 400 μm diameter solid pin as previously described [18]. Array hybridization procedures were performed as in our previous research and according to the manufacturer’s instructions [3,4]. A similar prepared array can detect the genomic DNA of pathogens at a concentration of 10 fg per nested PCR as described in our previous study [16].

The turnaround time of the whole process, from genomic DNA extraction to fungal hybridization results, was approximately 12 h, including two runs of PCR and signal detection using microarray. Once all the probes had been applied, the membrane was air-dried and exposed to shortwave UV (Stratalinker 1800; Stratagen, La Jolla, CA, USA) for 30 s. Unbound oligonucleotides were removed by two washes (2 min each) at room temperature in 0.5× SSC (1× SSC is 0.15 M NaCl plus 0.015 M sodium citrate)-0.1% sodium dodecyl sulfate (SDS). The arrays were stored at room temperature for further use [3].

Our previous studies demonstrated that our methods have a good ability to eliminate weak cross-hybridizations produced by nonhomologous species. With similar methods, our probes were intentionally designed to incorporate one single-base mismatch with their respective complementary target sequences to eliminate weak cross-hybridizations. Modified probes had no cross-hybridization with other species in previous studies [3,4].

### 2.4. Data Collection

During the study period (January 2017 to December 2020), patients suspected of having IMIs and undergoing tissue biopsy, concurrent with the PCR/microarray assay for the biopsied tissues, with or without tissue fungal culture, were enrolled for analysis. Clinical information was collected by reviewing electronic medical records, including demographic data, infectious diseases, and culture and histology reports, if available. The choices of antifungal therapy based on the available data of molecular detection, microbiological or histopathological studies, were evaluated at the discretion of the attending physicians.

### 2.5. Definition

The category of IMIs was determined according to the latest updated European Organization for Research and Treatment of Cancer and the Mycoses Study Group Education and Research Consortium (EORTC/MSGERC) consensus in 2020 [25]. Proven IMIs were determined based on either histopathological or culture evidence in the context of compatible infectious disease processes. A probable IMI refers to an invasive pulmonary infection, central nervous system infection, or sinonasal disease, requiring the presence of at least one host factor, a clinical feature, and mycological evidence for immunocompromised patients. Possible IMIs, on the other hand, meet only the criteria of a host factor and clinical feature without mycological evidence. Clinical suspicion of IMIs was defined as the presence of localized signs and symptoms that were attributed to mold infections without other recognized causes, not matching the EORTC/MSGERC categories, and treated with systemic antifungal agents for at least 10 days.

A positive galactomannan (GM) antigen result is defined as a galactomannan antigen index ≥ 1.0 in plasma/serum or bronchoalveolar lavage fluid or galactomannan antigen ≥ 0.7 in plasma/serum and ≥0.8 in bronchoalveolar lavage fluid, according to the updated criteria proposed by EORTC/MSGERC [25]. The GM level was determined by a Platelia *Aspergillus* Ag kit (BioRad^®^, Steenvoorde, France).

Positive culture results were defined as the pathogen isolated from the biopsied samples. The identification of hyphae in the same biopsied samples by special stains, such as Grocott’s methenamine silver or Periodic acid–Schiff staining, was defined as a positive histological finding. The immunocompromised status was determined based on the host factors defined in the EORTC/MSGERC criteria [25]. Malignancy included active hematological malignancy, solid tumors undergoing active treatment or stable patients receiving treatment before. Only chemotherapy against malignancies within 3 months prior to the tissue biopsy was recorded. Steroid therapy was classified as patients ever taking ≥ 15 mg prednisolone per day or other drugs with equivalent anti-inflammatory effects for at least 3 weeks before the biopsy procedure.

## 3. Results

### 3.1. Tissue Samples, Quality, and Mold Species Identified Using PCR/Microarray

During the study period, a total of 80 clinical samples from 74 patients were tested. The demographic and clinical characteristics of the enrolled patients are summarized (Table 1). The mean age of the patients was 52.0 years old, with males being predominant (59.5%). Malignancy and autoimmune diseases were the most commonly found underlying diseases. According to the immunocompromised host factors classified by the EORTC/MGS criteria, 25 (33.8%) were immunocompromised, predisposing them to mold infection. Five species of mold were identified by PCR/microarray assays (Table 2). *A. fumigatus* was the most common isolated species (N = 3). A total of 80 types of specimens and 8 positive PCR/microarray tissue samples, including lung/pleura (2), lymph nodes (2), bone/joint (1), liver (1), sinus (1), and middle ear tissue (1) were recorded (Table 3). Of the eight tissue samples with a positive result for PCR/microarray, three were categorized as proven IMI, one as probable IMI, one as possible IMI, two as clinical suspicion IMI, and one recovered uneventfully without receiving any antifungal therapy.

### 3.2. Comparison of PCR/Microarray Results, Histological Findings, Growth of Molds, and Diagnosis of IMIs in Different Types of Specimens

A diagnosis of an IMI was made in 10 patients (4 proven, 1 probable, 3 possible, 2 clinical suspicion) (Table 4). The PCR/microarray test was positive for three out of four proven IMIs, one probable IMI, and one out of three possible IMIs. Two patients with positive PCR/microarray findings, *A. fumigatus* for the hip joint, and *A. fumigatus* for the neck lymph node, were considered clinical suspicion IMIs, and the physicians responsible for those patients initiated antifungal therapy for them (voriconazole for the *A. fumigatus* lymphadenitis, and voriconazole for the *A. fumigatus* hip infection) despite no supporting microbiological and histological evidence. The other 70 samples were identified as non-IMI. Among them, one patient with a positive PCR/microarray result of *A. niger* for neck lymph nodes did not receive any antifungal therapy and recovered uneventfully. One male patient with a positive culture of *Phaeoacremonium* spp. from the knee joint but a negative PCR/microarray result was determined as a contaminated result, and the patient was not treated with any antifungal agents.

To compare the PCR/microarray results, the performance of the culture and histological analysis in diagnosing the IMIs are summarized (Table 5). The sensitivity and specificity of using PCR/microarray in diagnosing the IMIs were 75% and 95.7%, respectively. The positive predictive value (PPV) of PCR/microarray in making the diagnosis of the IMIs was 70%, and the negative predictive value (NPV) of this method was 95.7%. The sensitivity/specificity of culture and histological findings for diagnosing the IMIs were 10%/96.3% and 40%/100%, respectively. The PPV and NPV for fungal culture were 33.3% and 85.2%, respectively, and the PPV and NPV for histological analysis were 100% and 91.4%, respectively.

## 4. Discussion

In the present study, we validated the diagnostic performance of a PCR-based microarray for diagnosing invasive mold infections in tissue samples. This differs from previous studies that primarily focused on samples from BAL or plasma [14,15,26,27]. Among 80 infection sites, the microarray identified fungal pathogens in 3 proven, 1 probable, 1 possible, and 2 clinical-suspicion samples. Concordance between the microarray and traditional culture and histological analysis was 84.4% and 89.2%, respectively, which was similar with other broad-range ITS PCR methods for fungal detection [14,15,26,27]. The primary advantage of PCR methods on biopsies is to provide identification of the hyphae seen in pathology when the culture results are negative. Our results were compared with previous research conducted by Ala-Houhal et al. [28]. Their study demonstrated that fungal PCR could identify fungal pathogens in 59% of cases with proven or probable invasive fungal infections from deep tissues or fluids. Overall, they discovered that PCR findings were concordant (>86%) with conventional culture and histology.

Several commercial multiplex PCR tools have been developed for the identification of clinical IMIs [27]. Clinical data have proven that commercial multiplex PCR tools exhibit good performance in diagnosing invasive aspergillosis or mucormycosis [15,26,29]. Despite the advantages of rapidness and accuracy, the short diagnostic list for these targeted PCR methods may limit the identification of IMIs due to uncommon molds outside their diagnostic spectrum.

Compared with multiplex PCR tools, our method can identify a broader range of species at the same time. The mold array used in the present study covers 78 mold species commonly encountered in environments, and superficial and deep-seated infections. The majority of positive PCR results were *Aspergillus* species, with only one *Mucorales* identified, and no dermatophytes detected. These results suggest that our diagnostic procedures, following strict operating protocols, are free from contamination by environmental molds. The PCR/microarray test is comprised of *Fusarium* spp. and *Rhizopus*, which are molds commonly causing opportunistic infections in immunocompromised hosts in addition to *Aspergillus* spp. [30]. It is necessary to enroll more patients to validate its efficacy in diagnosing these rare IMIs in immunocompromised individuals.

Another advantage of microarrays is their ability to identify multiple mold species simultaneously, despite the scarcity of mixed invasive fungal infections. Conventional PCR is unable to determine DNA sequences precisely when it encounters specimens with mixed infections [31]. Sequencing of the amplicons could also identify the pathogens not included in the microarray if there is a strong clinical suspicion of fungal infection.

Metagenomic next-generation sequencing (mNGS) is a powerful culture-independent tool for identifying pathogens unbiasedly [32,33,34]. The benefit of mNGS is that there is no prerequisite for identification, which is a major advantage when the number of new molds responsible for infection is growing. However, most sequencing procedures of mNGS require a longer turnaround time compared to our microarrays. In addition, technical support and costs are concerns of the mNGS. In settings where mNGS is not available, the microarray assay could be a practical choice.

In this study, the sensitivity of the microarray for diagnosing IMIs was 70%, which was higher than that of the culture (10%) and histology (40%). For two patients with clinical suspicion of IMIs and who did not meet the EORTC/MSGERC criteria, the treating physicians initiated antifungal therapy based solely on positive microarray results, and the infections were cured in these cases. If these two clinically suspicious IMI cases are excluded according to the strict EORTC/MSGERC criteria, the sensitivity of microarray for diagnosing the IMIs would be 62.5%, 12.5% for culture, and 50% for histology. However, the PCR/microarray method can further determine molds at the species level, according to our previous research [3,4,16,17]. In the present, the PCR/microarray method provided additional microbiological evidence for three proven IMIs, one probable IMI, and one possible IMI. The tissue cultures only yielded positive results in the possible IMI case. The physicians responsible for those patients chose antifungal agents based on the identification of mold species or genus.

Additionally, the specificity of PCR in our study was not inferior to the culture and histology methods, with all methods showing a specificity of above 90%. The main differences between the PCR-based microarray and other detection methods, including serology and mNGS, were in terms of sensitivity, timeliness, cost, and technical requirements. Culture methods take a few days to weeks, and histology requires at least one week. Serology tests, such as the 1,3-β-d-glucan or *Aspergillus* galactomannan (GM) assay, have relatively lower sensitivity [35]. Metagenomic next-generation sequencing involves higher costs and technical requirements. Our in-house PCR assays demonstrated good sensitivity and specificity, comparable to several previously published studies [27]. Furthermore, our microarray is prepared in-house, reducing the time needed to send samples to a central lab. The cost for each test, which covers PCR and microarray production, is approximately USD 50, making it an affordable option for a standard laboratory compared to mNGS.

Although these two cases of clinical suspicion cannot be strictly considered as definitive diagnoses, they met the clinical criteria. However, we considered these two clinical suspicions to be underdiagnosed due to clinical improvement after treatment, likely related to challenges in culture cultivation. Furthermore, the duration of treatment remains inconclusive, with therapy based on the degree of clinical improvement, level of immunosuppression, and imaging response. In our study, Case 9 received approximately six weeks of treatment and demonstrated significant improvement; unfortunately, she did not continue follow-up in the internal medicine outpatient clinic. Case 10 underwent extended treatment lasting up to six months due to underlying autoimmune disease and the use of immunomodulators. Treatment was discontinued when no evidence of active pulmonary infection was found.

Determination of the species level promotes the precise use of antifungal agents. Our designed microarray can detect the most common *Aspergillus* species, *A. fumigatus*, *A. flavus*, *A. terreus*, and *A. niger* [36]. In the past decade, the incidence of azole-resistant *A. fumigatus* has increased [37,38]. In contrast, reduced susceptibility of *A. terreus* and *A. flavus* to amphotericin B has been recognized [39].

The main limitation of this study is that it is impossible to analyze the true sensitivity or specificity of these molecular diagnostic methods compared to a culture-based reference standard, especially for difficult-to-cultivate pathogens. Without positive histological evidence, any positive PCR results of biopsied tissue would be ambiguous. However, PCR-based diagnostic methods indeed shaped the clinical treatment when culture results and histological evidence are inconclusive. In our series, microorganisms were detected in two biopsy samples with positive PCR results, and both were considered as clinically suspicious for an IMI despite no supportive histological and culture evidence, indicating that the clinician had to rely on PCR results as the only microbiological evidence.

Another limitation is that the study included a very small number of patients. As a result, the calculated rates for sensitivity and specificity might be biased by the limited case numbers. Larger studies are needed to adequately describe the effectiveness of the method. In addition, contamination is still a limitation of all molecular diagnostic methods, even mNGS. Unlike RT-PCR, we cannot rely on Ct values for reference. With a negative control, we at least ruled out experimental contamination. To determine the presence of contamination during the sampling process would depend on combining the control with the physician’s judgment and the histopathological results to make the final decision.

## 5. Conclusions

In summary, our study highlights the utility of PCR-based diagnostic methods in tissue specimens, providing supportive microbiological evidence when culture results are inconclusive. Diagnosing invasive mold infections is often challenging due to diagnostic uncertainties, the relatively low sensitivity of culturing, and ambiguities in serology. The development of a reliable and practical diagnostic method remains a significant challenge. However, the combination of microarray with fungal culture and histology promotes the precise diagnosis of invasive mold infections, particularly in difficult-to-diagnose patients.

## Figures and Tables

**Table 1 jof-10-00530-t001:** Demographic and clinical features of the 74 patients who underwent tissue polymerase chain reaction and microarray analysis for suspected mold infections.

Patient Characteristics	No. (%)
Age, years (mean ± SD)	52.0 ± 20.1
Male	44 (59.5)
30-day mortality	5 (6.8)
Diabetes	17 (23.0)
Hypertension	14 (18.9)
Malignancy	24 (32.4)
Autoimmune diseases	18 (24.3)
HIV infection	4 (5.4)
Chronic kidney disease	8 (10.8)
End-stage renal disease with regular hemodialysis	4 (5.4)
Cardiovascular disease	18 (24.3)
Cirrhosis	2 (2.7)
Chemotherapy	9 (12.2)
Use of steroid	3 (4.1)
Immunocompromised status by EORTC/MSGERC criteria	25 (33.8)

EORTC/MSGERC, European Organization for Research and Treatment of Cancer and the Mycoses Study Group Education and Research Consortium.

**Table 2 jof-10-00530-t002:** Species of mold were identified by polymerase-chain reaction/microarray in the study.

Patient Characteristics
*Cunninghamella bertholletiae* (1)
*Aspergillus fumigatus* (3)
*Aspergillus niger* (1)
*Aspergillus terrus* (1)
*Aspergillus flavus* (2)

**Table 3 jof-10-00530-t003:** Polymerase-chain reaction (PCR) and microarray findings over different tissue samples.

Tissue Samples (Total No.)	No. (%) of Positive PCR Result	Molds Based on the PCR Result/MI Category
Bone and joint (20)	1 (5)	*Aspergillus fumigatus*/clinical suspicion
Lung/pleura (17)	2 (11.8)	*Cunninghamella bertholletiae*/proven and *Aspergillus niger*/probable
Lymph node (15)	2 (13.3)	*Aspergillus fumigatus*/clinical suspicion;
		*Aspergillus niger*/contamination
Valve/pericardium/aorta (5)	0	
Liver (4)	1 (25)	*Aspergillus fumigatus*/proven
Bone marrow (4)	0	
Sinus (3)	1 (33.3)	*Aspergillus flavus*/proven
Middle ear (2)	1 (50)	*Aspergillus flavus*/possible
Others (10)	0	
Total (80)	8 (10.0)	3 proven, 1 probable, 1 possible, 2 clinical suspicion, 1 contamination

MI, mold infection.

**Table 4 jof-10-00530-t004:** Clinical aspects of 10 patients with invasive mold infections (IMI). The categories of MIs include proven, probable, and possible invasive mold infections, as well as clinical suspicion.

Age */Sex	Specimen/Biopsy Method	Category of IMI	Underlying Diseases	Hyphae Observed in Histological Finding	Fungal Culture	Microarray Result	Treatment Adjusted by Microarray Finding	Primary Antifungal Treatment	Outcome
57/M	Lung/CT-guide biopsy	Proven	AML	Yes	No growth	*Cunninghamella bertholletiae*	Yes	Posaconazole	Expired
72/M	Liver/sono-guide biopsy	Proven	HHV-8-associated multicentric Castleman disease, Kaposi’s sarcoma	Yes	Not done	*Aspergillus fumigatus*	No	Amphotericin B, voriconazole, posaconazole	Under control with chronic suppressive therapy
52/F	Paranasal sinus/surgical biopsy	Proven	MDS, haplo-identical allogeneic peripheral blood stem cell transplantation, GVHD	Yes	No growth	*Aspergillus flavus*	No	Amphotericin B, voriconazole	Expired
60/M	Lung/surgical biopsy	Proven	Chronic kidney disease, cirrhosis, aortic dissection	Yes	No growth	Negative	No	Liposomal amphotericin B	Expired
63/M	Lung/CT-guide biopsy	Probable	AML	No	No growth; serum GM:2.6 (positive)	*Aspergillus niger*	No	Voriconazole	Cured
89/F	Middle ear/surgical biopsy	Possible	ESRD, DM, HTN, Alzheimer’s diseases	No	*Aspergillus* species	*Aspergillus flavus*	No	Voriconazole	Expired
80/F	Para-nasal sinus/surgical biopsy	Possible	Nasal lymphoma, lung adenocarcinoma	No	No growth	Negative	No	Liposomal amphotericin B, voriconazole	Cured
18/M	Lung/CT-guided biopsy	Possible	AML, chronic GVHD	No	No growth	Negative	No	Voriconazole	Cured
61/F	Hip joint/surgical biopsy	Clinical suspicion	End-stage renal disease, DM aortic stenosis	No	No growth	*Aspergillus fumigatus*	Yes	Voriconazole	Cured
63/F	Lymph node/surgical biopsy	Clinical suspicion	Sjogren syndrome	No	No growth	*A. fumigatus*	Yes	Voriconazole	Cured

*, year-old; CT, computed tomography; AML, acute myeloid leukemia; DM, diabetes mellitus; HTN, hypertension; MDS, myelodysplastic syndrome; GVHD, graft versus host-disease; ESRD, end-stage renal disease; GM, Galactomannan test.

**Table 5 jof-10-00530-t005:** Comparison sensitivity and specificity of polymerase chain reaction/microarray with culture and histological findings in the diagnosis of 10 invasive mold infections (IMI). The IMI group includes proven, probable, possible, and clinical suspicion IMI.

	Number of Positive Patients/Sensitivity (%)		
	PCR	Culture	Histology
IMI (n = 10)	7/70%	1/10%	4/40%
Proven (n = 4)	3/75%	0/0%	4/100%
Probable (n = 1)	1/100%	0/0%	0/0%
Possible (n = 3)	1/33.3%	1/33.3%	0/0%
Clinical suspicion IMI (n = 2)	2/100%	0/0%	0/0%
	Number of Positive Patients/Specificity (%)		
Non-IMI (n = 64)	3/95.7%	2/96.3%	0/100%

## Data Availability

The original contributions presented in the study are included in the article/Appendix A, further inquiries can be directed to the corresponding author.

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
