# Peer review of "PCR-Based Microarray Enhances Diagnosis of Culture-Negative Biopsied Tissue in Patients with Invasive Mold Infections: Real-World Experience in a Tertiary Medical Center"

_jof, 2024, doi:10.3390/jof10080530_

Round 1

Reviewer 1 Report

In this manuscriptmicroarray analysis was performed on biopsied tissues of 80 clinical samples from 74 patients with suspected IMI. The results indicated the sensitivity and specificity of PCR-based microarray were higher, which would provide supportive microbiological evidence when culture results are inconclusive. This method combined with fungal culture and histology will contribute to the accurate diagnosis of IMI in difficult-to-diagnose patients, which is of significance for the early diagnosis and treatment of IMI.

1.       In this study, two PCR reactions were used to complete the subsequent microarray detection, which took a total of 12 hours. Whether the detection be done by one PCR reaction?

2.       When the culture result and the histopathological examination are negative, and the PCR microarray result is positive, should IMI be diagnosed? How should the treatment be done?    The related statements might be added in the discussion section.

3.       It is suggested that the differences in specificity, sensitivity and timeliness between the PCR-based microarray and other detected methods should be compared in the discussion.

 In references, the names of microorganisms need to be italicized.

Author Response

Comments 1: In this study, two PCR reactions were used to complete the subsequent microarray detection, which took a total of 12 hours. Whether the detection be done by one PCR reaction?

Response 1: We are thankful for the reviewer’s query. It takes around a total of 12 hours to complete the whole process, including two runs of PCR and signal detection using microarray. To clarify this detail, the revised manuscript is as follows.

Page 3, Line 125-127: The turnaround time of whole process from genomic DNA extraction to fungal hybridization results is approximately 12 hours, including two runs of PCR and signal detection using microarray.

Comments 2: When the culture result and the histopathological examination are negative, and the PCR microarray result is positive, should IMI be diagnosed? How should the treatment be done? The related statements might be added in the discussion section.

Response 2: We are thankful for the reviewer’s query. We should emphasize this explanation in the discussion section. According to the latest updated European Organization for Research and Treatment of Cancer and the Mycoses Study Group Education and Research Consortium (EORTC/MSGERC) consensus in 2020, the clinical diagnosis of IMI based on host factor, clinical feature and mycological criteria including PCR. If the culture and histopathological examination were negative, and the PCR microarray result was positive, probable or possible IMI should be diagnosed only host factors and clinical features or either was met. Two clinical suspicions only met the clinical criteria so they were not strictly diagnosed as IMI according to the criteria. However, we considered these two clinical suspicions to be underdiagnosed due to clinical improvement after treatment, likely related to the challenges of culture cultivation. The duration of treatment was based on the degree of clinical improvement, level of immunosuppression, and imaging response. Case 9 received approximately six weeks of treatment and showed significant improvement; unfortunately, she did not continue follow-up in the internal medicine outpatient clinic. Case 10 underwent an extended treatment duration of up to six months due to underlying autoimmune disease and the use of immunomodulators. Treatment was discontinued until no evidence of active pulmonary infection. We focused on these explanations in the discussion section. The revised text is as follows.

Page 9, Line 293-303: Although these two cases of clinical suspicion cannot be strictly considered definitive diagnoses, they met the clinical criteria. However, we considered these two clinical suspicions to be underdiagnosed due to clinical improvement after treatment, likely related to challenges in culture cultivation. Furthermore, the duration of treatment remains inconclusive, with therapy based on the degree of clinical improvement, level of immunosuppression, and imaging response. In our study, Case 9 received approximately six weeks of treatment and demonstrated significant improvement; unfortunately, she did not continue follow-up in the internal medicine outpatient clinic. Case 10 underwent extended treatment lasting up to six months due to underlying autoimmune disease and the use of immunomodulators. Treatment was discontinued until no evidence of active pulmonary infection.

Comments 3: It is suggested that the differences in specificity, sensitivity and timeliness between the PCR-based microarray and other detected methods should be compared in the discussion.

Response 3: We are thankful for the reviewer’s query. We mentioned it’s difficult to understand the true sensitivity or specificity of these molecular diagnostic methods and compared to a culture-based reference standard (Line 274). We already described the sensitivity of the microarray for diagnosing IMI was 70%, higher than that of culture (10%) and histology (40%) in Line 257. We revised our text and mentioned additional specificity and timeliness in the text in the discussion to emphasize this point. The modified text is as follows.

Page 9, Line 280-291 & Reference 36: Additionally, the specificity of PCR in our study was not inferior to culture and histology, with all methods showing specificity above 90%. The main differences between the PCR-based microarray and other detection methods, including serology and mNGS, were in sensitivity, timeliness, cost, and technical requirements. Culture methods take a few days to weeks, and histology requires at least one week. Serology tests, such as the 1,3-β-d-glucan or Aspergillus galactomannan (GM) assay, have relative lower sensitivity [36]. Metagenomic next-generation sequencing involves higher costs and technical requirements. Our in-house PCR assays demonstrated good sensitivity and specificity, comparable to several previously published studies [27]. Furthermore, our microarray is prepared in-house, reducing the time needed to send samples to a central lab. The cost for each test, which covers PCR and microarray production, is approximately fifty US dollars, making it an affordable option for a standard laboratory compared to mNGS.

Comments 4: In references, the names of microorganisms need to be italicized.

Response 4: The mistakes in references were corrected.

Reviewer 2 Report

Thank you for providing me the opportunity to review this manuscript. It is interesting and well-written. I have some comments that could be of use:

1)      Line 154: I find this sentence hard to understand. I think it needs to be revised. What does simultaneously imply here?

2)      Notably, in Table 1, the proportion of those on chemotherapy was less than half of those with malignancy. This is strange, since having a diagnosis of malignancy, based on the definition in the last paragraph of the methods section, implies active treatment. How is that explained?

3)      The authors correctly identify the difficulty to calculate the true sensitivity of the method for the reasons they state. Another very important limitation is that the study includes a very small number of patients, thus, the calculated rates for sensitivity, specificity, etc., are merely rough approximations. Larger studies are needed to adequately describe the effectiveness of the method

Author Response

Comments 1: Line 154: I find this sentence hard to understand. I think it needs to be revised. What does simultaneously imply here?

Response 1: The revised text is as follows.

Page 4, Line 164: Positive culture results were defined as the pathogen isolated from the biopsied samples.

Comments 2: Notably, in Table 1, the proportion of those on chemotherapy was less than half of those with malignancy. This is strange, since having a diagnosis of malignancy, based on the definition in the last paragraph of the methods section, implies active treatment. How is that explained?

Response 2: We are thankful for the reviewer’s comments. We had revised descriptions to make it clearer. The malignancy category includes both active malignancies under treatment and stable patients who have already received treatment and are in complete or partial remission. Chemotherapy is defined as treatment received within the 3 months prior to the tissue biopsy in Line 159, which may indicate the status of immunocompromise. Hence, the proportion of those on chemotherapy was less than half of those with malignancy. We have revised the text to clarify the definition of malignancy more explicitly. The revised manuscript is as follows.

Page 4, Line 168-171: Malignancy included active hematological malignancy, solid tumors undergoing active treatment or stable patients receiving treatment before. Only chemotherapy against malignancies within 3 months prior to the tissue biopsy was recorded.

Comments 3: The authors correctly identify the difficulty to calculate the true sensitivity of the method for the reasons they state. Another very important limitation is that the study includes a very small number of patients, thus, the calculated rates for sensitivity, specificity, etc., are merely rough approximations. Larger studies are needed to adequately describe the effectiveness of the method.

Response 3: We are grateful of the reviewer’s suggestions. We agree and have addressed this issue in the discussion section. The revised manuscript is as follows.

Page 10, Line 318-321: Another limitation is that the study included a very small number of patients. As a result, the calculated rates for sensitivity and specificity might be biased by limited case numbers. Larger studies are needed to adequately describe the effectiveness of the method.

Reviewer 3 Report

This publication is interesting, the proposed system can undoubtedly influence the development of IMI diagnostics. This system has many advantages, although no one paid attention to the system's disadvantages.

The presented results are promising and the method seems to have great potential. However, for the method to be implemented as a diagnostic method, many tests must be carried out. For this reason, you should carefully describe the preparation of the entire procedure from scratch and demonstrate the durability of the prepared flattened tiles. Time and costs also influence this. Of course, I'm not talking about the exact cost, but a comparison of the costs of different methods. And what I mentioned earlier about testing DNA samples to exclude false positive results.

Author Response

Comments 1: Have tests been carried out with many samples to exclude false positive results, i.e. whether there are any cross-reactions between other species of fungi, bacteria, and others? What is the risk of contamination with PCR products or fungi found in the environment? Are any precautions taken? We should also check the specificity if we have an infection with several species. You can also check whether the amount of DNA limits the negative results for tests in which fungal DNA is present (false negative). It is also not described in detail how Microarray tiles are prepared for flattening, how they are stored, and how long they are useful for use.

Response 1: We thank the reviewer’s query to clarify our methods. Contamination can be divided into sample contamination and procedural contamination. During the sampling process, we use sterile methods (intraoperative and biopsy) to minimize contamination. Each microarray includes a negative control to ensure that there is no contamination during the experimental process.

As for cross reactions, our previous studies demonstrated our methods have a good ability to eliminate weak cross-hybridizations produced by nonhomologous species. We produced the probes using the similar methods described in a previous study conducted by Leaw et al. [4]. These probes were intentionally designed to incorporate one single base mismatch with their respective complementary target sequences to eliminate weak cross-hybridizations. Modified probes had no cross-hybridization with other species, but still displayed good hybridization signals with their respective target yeasts. Leaw et el. tested total 309 target strains and concluded that almost all the strains were correctly identified by the oligonucleotide array, producing a test sensitivity of 100% and there were no any hybridization signals with probes on the array except the positive control probe. The test specificity of the array was 97% (32/33). Similarly, in another previous study, Hung et al. [17], no cross-hybridization was observed for the tested strains and a specificity of 100% (66/66) was achieved.

As mentioned in previous study, Hsiao et al. [3], detection limit was the lowest amount of fungal DNA that could be detected by the array. Our method cannot quantify DNA because it uses a fixed number of reaction cycles. Serial 10-fold dilutions of DNAs of Aspergillus fumigatus BCRC 30502 and A. versicolor BCRC 31488 were used to determine the detection limits.

With the same storage and fabrication, once all probes had been applied, the membrane was air-dried and exposed to shortwave UV (Stratalinker 1800; Stratagen, La Jolla, Calf.) for 30 s. Unbound oligonucleotides were removed by two washes (2 min each) at room temperature in 0.5× SSC (1× SSC is 0.15 M NaCl plus 0.015 M sodium citrate)-0.1% sodium dodecyl sulfate (SDS). The arrays were stored at room temperature for further use. Other detail was described in paragraph of Fabrication of membrane assays and hybridization procedures. Microarrays are prepared in batches of 100, stored properly in a desiccator, and used within six months. Each microarray includes a positive control to ensure its effectiveness each time it is used. The revised manuscript is addressed together in response to Comments 3.

Comments 2: There was no mention of the financial costs of such use and the equipment required for this purpose. Cost often influences the lack of routine use of some diagnostic solutions. Is 12 hours the total time from sample collection to testing?

Response 2: We are thankful for the reviewer’s comments. The financial cost is low. Our microarray is prepared in-house, reducing the time needed to send samples to a central laboratory. The cost of PCR and microarray production is affordable for a standard laboratory, especially when compared to mNGS. Our cost for each test is approximately fifty US dollars. As for time, it takes around a total of 12 hours to complete the whole process, including two runs of PCR and signal detection using microarray. The manuscript is revised as follows.

Page 9, Line 286-291: Metagenomic next-generation sequencing involves higher costs and technical requirements. Our in-house PCR assays demonstrated good sensitivity and specificity, comparable to several previously published studies [27]. Furthermore, our microarray is prepared in-house, reducing the time needed to send samples to a central lab. The cost for each test, which covers PCR and microarray production, is approximately fifty US dollars, making it an affordable option for a standard laboratory compared to mNGS.

Page 3, Line 125-127: The turnaround time of whole process from genomic DNA extraction to fungal hybridization results is approximately 12 hours, including two runs of PCR and signal detection using microarray.

Comments 3: The presented results are promising and the method seems to have great potential. However, for the method to be implemented as a diagnostic method, many tests must be carried out. For this reason, you should carefully describe the preparation of the entire procedure from scratch and demonstrate the durability of the prepared flattened tiles. Time and costs also influence this. Of course, I'm not talking about the exact cost, but a comparison of the costs of different methods. And what I mentioned earlier about testing DNA samples to exclude false positive results.

Response 3: We are thankful for the reviewer’s comments and suggestions. We agree the time and costs influence the applicability and feasibility, which was mentioned above in response 2. We add a paragraph to display the issue of contamination, storage, cross reactions to exclude false positive results as possible. We can rule out experimental contamination because we have negative controls, but contamination during the sampling process depends on whether findings match histological results, combined with the physician's judgment to make the final decision. Unlike RT-PCR, we cannot rely on Ct values for reference, but the limitation of all molecular diagnostic methods, including mNGS, is contamination at any step of the procedures. The manuscript is revised as follows.

Page 10, Line 321-325: Besides, contamination is still a limitation of all molecular diagnostic methods even mNGS. Unlike RT-PCR, we cannot rely on Ct values for reference. With negative control, we at least ruled out experimental contamination. To determine the contamination during the sampling process depends on combining it with the physician's judgment and histopathological results to make the final decision.

Page 3, Line 127-131: Once all probes had been applied, the membrane was air-dried and exposed to shortwave UV (Stratalinker 1800; Stratagen, La Jolla, Calf.) for 30 s. Unbound oligonucleotides were removed by two washes (2 min each) at room temperature in 0.5× SSC (1× SSC is 0.15 M NaCl plus 0.015 M sodium citrate)-0.1% sodium dodecyl sulfate (SDS). The arrays were stored at room temperature for further use [3].

Page 3, Line 132-136: Our previous studies demonstrated our methods have a good ability to eliminate weak cross-hybridizations produced by nonhomologous species. With the similar methods, our probes were intentionally designed to incorporate one single base mismatch with their respective complementary target sequences to eliminate weak cross-hybridizations. Modified probes had no cross-hybridization with other species in previous studies [3,4].

Round 2

Reviewer 2 Report

The manuscript has been improved.

None

Reviewer 3 Report

  I do not have any remarks.

  Thank you for your comprehensive justification. I am waiting for further reports analyzing a larger number of samples